# The New Era of TIRADSs to Stratify the Risk of Malignancy of Thyroid Nodules: Strengths, Weaknesses and Pitfalls

**DOI:** 10.3390/cancers13174316

**Published:** 2021-08-26

**Authors:** Gilles Russ, Pierpaolo Trimboli, Camille Buffet

**Affiliations:** 1Groupe de Recherche Clinique n°16 Tumeurs Thyroïdiennes, Thyroid and Endocrine Tumors Unit, Institute of Endocrinology, Pitié-Salpêtrière Hospital, Sorbonne University, F-75013 Paris, France; camille.buffet@aphp.fr; 2Clinic for Endocrinology and Diabetology, Lugano Regional Hospital, Ente Ospedaliero Cantonale, 6900 Lugano, Switzerland; Pierpaolo.Trimboli@eoc.ch; 3Faculty of Biomedical Sciences, Università della Svizzera Italiana (USI), 6900 Lugano, Switzerland

**Keywords:** thyroid, nodule, risk stratification, TI-RADS, fine-needle aspiration

## Abstract

**Simple Summary:**

The aim of this review is to provide the reader with a comprehensive overview of thyroid imaging and reporting data systems used for thyroid nodules, so as to understand how nodules are scored with all existing systems. Both ultrasound based risk stratification systems and indications for fine-needle aspirations are described. Systems are compared by analyzing their strengths and weaknesses. Studies show satisfactory sensitivities and specificities for the diagnosis of malignancy for all systems, and none of them have shown a real significant advantage over the others in terms of raw diagnostic value. Interobserver agreement is also very similar for all systems, fairly adequate to robust. Dimensional cut-offs for fine-needle aspiration are quite similar and all RSSs seem to reduce effectively the number of unnecessary FNAs. Merging all existing systems in a common international one is desirable.

**Abstract:**

Since 2009, thyroid imaging reporting and data systems (TI-RADS) have been playing an increasing role in the field of thyroid nodules (TN) imaging. Their common aims are to provide sonologists of varied medical specialties and clinicians with an ultrasound (US) based malignancy risk stratification score and to guide decision making of fine-needle aspiration (FNA). Schematically, all TI-RADSs scores can be classified as either pattern-based or point-based approaches. The main strengths of these systems are their ability (i) to homogenize US TN descriptions among operators, (ii) to facilitate and shorten communication on the malignancy risk of TN between sonologists and clinicians, (iii) to provide quantitative ranges of malignancy risk assessment with high sensitivity and negative predictive values, and (iv) to reduce the number of unnecessary FNAs. Their weaknesses are (i) the remaining inter-observer discrepancies and (ii) their insufficient sensitivity for the diagnosis of follicular cancers and follicular variant of papillary cancers. Most common pitfalls are degenerating shrinking nodules and confusion between individual and coalescent nodules. The benefits of all TI-RADSs far outweigh their shortcomings, explaining their rising use, but the necessity to improve and merge the different existing systems remains.

## 1. Introduction

Risk stratification systems (RSSs) have two main aims. The first one is to homogenize the results of thyroid ultrasound (US) reports, by using a quantitative cancer risk estimation approach, in order to facilitate communication between practitioners and with the patients. Ambiguities of qualitative descriptions such as “multinodular goiter to be confronted with biological tests” are reduced and allow for a quick understanding of the risk level of a thyroid nodule. The second one is to provide guidelines regarding the indications for fine-needle aspiration biopsy (FNA). There again, the limitation of subjectivity for this decision is crucial for patients to hope to get homogenized care.

Some of these systems, but not all, have incorporated a lexicon and even more rarely a standardized report. At least the former seems mandatory to increase inter-observer description agreement.

However, all RSSs tend to base the whole stratification and decision making process solely on US criteria and nodular size, whereas obviously many other factors should, and are, integrated when accomplishing these tasks. Among these are patient’s age and sex; age of the disease; family history of thyroid cancer; personal history of cervical irradiation; clinical symptoms such as dysphonia, dysphagia, or dyspnea; nodular location; number of nodules; and presence of suspicious cervical lymph nodes. Thus, a more thorough algorithm, also including laboratory tests such as TSH and calcitonin and thyroid scintigraphy when deemed adapted, may be sought in the future. This review will describe present RSSs, their strengths, weaknesses, and pitfalls via a comprehensive analysis of the literature and make some suggestions for the future.

### 1.1. Description of Present RSSs

Several national and international professional organizations have developed US-based risk-stratification systems. They are often referred to as thyroid imaging reporting and data systems, or TIRADS, terms derived from those used for breast cancer imaging. Some societies have chosen to stay with their own name to refer to their system (e.g., the American Thyroid Association). RSSs assign thyroid nodules to categories characterized by increasing risk ranges for cancer, based on the presence or not of specific US features. Two of the eight RSSs described below, ACR- and C-TIRADS, are point-based systems and the six others are pattern-based. Pattern-based scoring consists of recognizing a grouping of US features in a single figure, whereas point-based scoring systems consist of summing points that have been formerly attributed to US features.

#### 1.1.1. Chilean TIRADS (2009)

Historically, it was the first TIRADS to be published [1]. Ten US patterns were defined, called colloid 1 to 3 (TIRADS 2), pseudo-nodule (TIRADS 3), simple neoplastic, De Quervain and suspicious neoplastic patterns (TIRADS 4A), malignant A (TIRADS 4B), B, and C patterns (TIRADS 5). TIRADS 2 corresponded to anechoic with hyperechoic spots, nonvascularized lesions, or to nonencapsulated, mixed isoechoic with hyperechoic spots lesions and to spongiform nodules. TIRADS 3 nodules referred to hyper, iso, or hypoechoic, partially encapsulated nodules with peripheral vascularization, in Hashimoto’s thyroiditis. TIRADS 4A nodules were solid or mixed hyper, iso, or hypoechoic nodules, with a thin capsule, or hypoechoic lesion with ill-defined borders but without calcifications or hyper, iso, or hypoechoic, hypervascularized, encapsulated nodules with a thick capsule, containing calcifications (coarse or microcalcifications). TIRADS 4B corresponded to hypoechoic, nonencapsulated nodules, with irregular shape and margins, penetrating vessels, and with or without calcifications, and TIRADS 5 referred to iso or hypoechoic, nonencapsulated nodules with multiple peripheral microcalcifications and hypervascularization or nonencapsulated, isoechoic mixed hypervascularized nodules with or without calcifications, without hyperechoic spots. The TIRADS classification was evaluated in a sample of 1097 nodules (benign: 703; follicular lesions: 238; and carcinoma: 156), among which all nodules with a malignant FNAB result were submitted to surgery, benign ones by FNAB were followed, and in the group of patients with indeterminate or follicular lesions, 31% were operated on, and the rest followed. Sensitivity, specificity, positive predictive value (PPV), negative predictive value (NPV), and accuracy were 88, 49, 49, 88, and 94%, respectively.

In 2016, in a study on 210 patients with 502 nodules, the same team found a 99.6% sensitivity, a 74.35% specificity, an 82.1% PPV, and a 99.4% NPV [2].

#### 1.1.2. BTA Classification (British Thyroid Association) (2014)

In 2014 the British Thyroid Association guidelines for the management of thyroid cancer were introduced [3]. The BTA system classifies the thyroid US features in 5 categories at increasing risk of malignancy, from U1 (normal thyroid gland) to U5 (very suspicious lesion). The U2 (benign) category is characterized by isoechoic or mildly hyperechoic nodules with halo, cystic change with or without “ring down sign,” microcystic or spongiform appearance, peripheral eggshell calcification, or peripheral vascularity. The U3 (indeterminate/equivocal) category comprises homogeneous, hyperechoic (markedly), solid nodules with halo (follicular lesion); hypoechoic nodules with equivocal echogenic foci cystic change; or mixed/central vascularity. The U4 (suspicious) category is characterized by solid hypoechoic or very hypoechoic nodules with disrupted peripheral calcification and hypoechoic lobulated outline. The U5 (malignant) category comprises solid hypoechoic nodules with lobulated or irregular outline and with or without microcalcification or globular calcification. Other U5 malignant features are intranodular vascularity, taller-than-wide shape, and characteristic associated lymphadenopathy.

In 2020, a retrospective observational study was carried out among 1465 patients. Thyroid surgery was performed in 129 patients, of which malignancy was seen in 35 (27.1%). The proportions of patients with cancer in U1–U5 categories were 0%, 13.6%, 30.4%, 40%, and 100%, respectively [4]. In another study of 73 consecutive patients with 17 histological confirmed malignant nodules, it was found that the sensitivity and NPV of BTA-U score in detecting and predicting malignancy were 100%, whereas the specificity and PPV were 34% and 32%, respectively [5].

#### 1.1.3. AACE (American Association of Clinical Endocrinologists) Grading System (2016)

In 2016 the American Association of Clinical Endocrinologists (AACE), American College of Endocrinology (ACE) and Associazione Medici Endocrinologi (AME) Medical Guidelines for Clinical Practice for the Diagnosis and Management of Thyroid Nodules [6] were released. These included recommendations on reporting, an illustrated atlas, and an assessment of the malignancy risk of all US features, including Doppler and elastography and suggested a 3-tier RSS subdivided into low, intermediate, and high risks. Low risk nodules corresponded to cystic and spongiform ones and intermediate risk nodules to mildly hypoechoic and isoechoic ones with no features of high suspicion. The latter included marked hypoechogenicity, spiculated or lobulated margins, microcalcifications, taller-than-wide shape, extrathyroidal growth, and/or a pathologic lymph node.

In 2017, a study on 859 FNAs from 598 patients showed that 88.5% and 74.9% of low and intermediate risk nodules, respectively, were cytologically benign, whereas 84.6% of high risk nodules had a moderate-to-elevated risk of malignancy or were malignant [7].

#### 1.1.4. ATA (American Thyroid Association) Grading System (2016)

In 2016, the American Thyroid Association Management Guidelines for Adult Patients with Thyroid Nodules and Differentiated Thyroid Cancer were published [8]. The RSS was composed of 5 categories ranging from benign to high suspicion. Irregular margins (infiltrative, microlobulated), microcalcifications, taller than wide shape, rim calcifications with small extrusive soft tissue component, and evidence of extra-thyroidal extension were considered highly suspicious in hypoechoic nodules, and, on the contrary, cystic nodules were classified as benign and spongiform nodules as very low suspicion. Low suspicion and intermediate suspicion nodules depended on their echogenicity and composition (hypoechoic solid-intermediate suspicion and isoechoic solid or partially cystic-low suspicion). The ATA risk assessment was validated in a prospective study on 206 nodules [9]. Malignancy rates determined by cytology/surgical pathology were 100%, 11%, 8%, and 2% in high, intermediate, low, and very low classes, respectively, which were closely aligned with ATA malignancy risk estimates (high 70–90%, intermediate 10–20%, low 5–10%, and very low 3%).

#### 1.1.5. K-TIRADS (Korean-TIRADS) (2016)

The last published version of the Korean Society of Thyroid Radiology was issued in 2016 [10], including a detailed lexicon, an RSS, and management recommendations. The main specificities of the lexicon were to detail definitions of composition and of hyperechoic foci. Composition can be solid, predominantly solid, or cystic (with a 50% cut-off) or (entirely) cystic (meaning no solid portion). Hyperechoic foci can correspond either to microcalcifications when measuring 1 mm or less and located in the solid portion, or to colloid when located in the cystic portion and generating comet-tail artifacts. The RSS ranges from 1 to 5, 1 corresponding to the absence of nodule. It is based on both composition and echogenicity. Pure cysts and spongiform nodules are scored as K-TIRADS 2 (benign). Iso/hyperechoic nodules and partially cystic nodules are classified as K-TIRADS 3 (low suspicion) in the absence of features of high suspicion and as K-TIRADS 4 (intermediate suspicion) if there is any suspect feature, likewise solid hypoechoic nodules without features of high suspicion. Solid hypoechoic nodules with any suspicious features (microcalcification, nonparallel orientation, spiculated/microlobulated margins) are K-TIRADS 5 (high suspicion).

Its diagnostic value has been evaluated in a prospective multicenter study on 902 nodules [11]. The calculated malignancy risk in K-TIRADS categories 5, 4, 3, and 2 nodules was 73.4, 19.0, 3.5, and 0.0%, respectively. The sensitivity, specificity, PPV, NPV, and accuracy for malignancy were 95.5, 58.6, 44.5, 96.9, and 69.5%, respectively.

#### 1.1.6. EU-TIRADS (European-TIRADS) (2017)

Published in 2017, the European Thyroid Association (ETA) guidelines include a lexicon, a standardized report, an RSS, and management recommendations [12]. The lexicon incorporates illustrations and the report a drawing example used to locate nodules simply and precisely. The RSS ranges from 1 to 5, with 1 corresponding to no nodule. EU-TIRADS 2 correspond to purely cystic and spongiform nodules. EU-TIRADS 3 are isoechoic nodules with no features of high suspicion and EU-TIRADS 4 mildly hypoechoic nodules also with no such features, knowing that here the presence of a mildly hypoechoic zone, even in minority, is sufficient to classify the nodule as intermediate risk. Features of high suspicion are marked hypoechogenicity, microcalcifications, taller-than-wide shape, spiculated/microlobulated margins, and the presence of at least one of these categorizes the nodule as EU-TIRADS 5.

A multicenter retrospective validation study on 1058 nodules using final histology as a gold standard found a cancer rate within or close to the given range described in the EU-TIRADS guidelines and a satisfactory diagnostic value with 93% sensitivity and 97% NPV [13]. A meta-analysis published in 2020 including seven studies and evaluating 5672 nodules showed that the prevalence of malignancy in each EU-TIRADS class was 0.5%, 5.9%, 21.4%, and 76.1%, from class 2 to 5 respectively. The sensitivity, specificity, PPV, and NPV of EU-TIRADS class 5 for the detection of malignancy were 83.5%, 84.3%, 76.1%, and 85.4%, respectively [14].

#### 1.1.7. ACR-TIRADS (American College of Radiology-TIRADS) (2017)

The lexicon was issued in 2015 [15] and the RSS and management recommendations in 2017 [16]. In contrast to most other RSSs, the ACR-TIRADS is point-based, considering five US categories, which are composition, echogenicity, shape, margin, and echogenic foci. In each category, US features are attributed a certain number of points ranging from 0 to 3. Summing the points allows one to obtain the final classification of the nodule, which goes from 1 to 5, with 1 corresponding to benign, 2 to not suspicious, 3 to mildly suspicious, 4 to moderately suspicious, and 5 to highly suspicious. Features attributing 1 point are mixed composition, isoechogenicity, and macrocalcifications. Solid composition, hypoechogenicity, irregular margins, and peripheral calcifications correspond to 2 points. Marked hypoechogenicity, a taller-than-wide shape, extra-thyroidal extension, and all punctate echogenic foci give 3 points.

In a retrospective study on 100 nodules, sensitivity, specificity, and accuracy were 92% (95% CI: 68%, 98%), 44% (95% CI: 33%, 56%), and 52% (95% CI: 40%, 63%), respectively [17]. In a multi-institutional study aiming to analyze thyroid nodule risk stratification on 3422 nodules including 352 carcinomas, 2948 (86.1%) had risk levels that were within 1% of the TIRADS risk thresholds defined in the guidelines. Of the 474 nodules that were more than 1% outside these thresholds, 88.0% (417/474) had a risk level that was below the TIRADS threshold [18]. In a systematic review and meta-analysis on 31,552 nodules, the pooled sensitivity and specificity were 89% (95% CI 81–93%) and 70% (95% CI 60–78%), respectively. The calculated area under summary ROC was 0.86 (95% CI 0.83–0.89) [19].

#### 1.1.8. C-TIRADS (Chinese-TIRADS) (2020)

Realizing that in China, as many as ten versions of TIRADS had been used in different hospitals nationwide, causing a lot of confusion, the Chinese-TIRADS, in line with China’s national conditions and medical status, was established based on literature review, expert consensus, and multicenter data provided by the Chinese Artificial Intelligence Alliance for Thyroid and Breast Ultrasound [20]. It includes a terminology section and a score. The score ranges from 1 to 5, 1 corresponding to the absence of nodule. Each US feature is attributed a number of points ranging from −1 to 1 and the points are summed. Vertical orientation, solid composition, markedly hypoechoic, microcalcifications, ill-defined and irregular margins, and extra-thyroidal extension each are attributed 1 point, whereas comet-tail artifacts correspond to −1. The sum corresponds to the C-TIRADS score: 1, no nodule; 2, benign (−1 point); 3, probably benign (0 point); 4A, low suspicion (1 point); 4B, moderate suspicion (2 points); 4C, high suspicion (3–4 points); 5, highly suggestive of malignancy (5 points). The corresponding expected malignancy risks are 0, 0, ≤2, 2–10, 10–50, 50–90, and ≥90%, respectively. C-TIRADS 6 corresponds to a proven malignancy.

A multicentric retrospective validation study on 2141 thyroid nodules that were neither cystic nor spongiform was simultaneously published [21]. It was designed to determine which of three methods, namely regression equation, weighting, and counting would be the most suitable to determine the malignant risk of thyroid nodules. The counting value of positive and negative ultrasound features was retained to define the C-TIRADS. The malignancy risk of each TIRADS score was in agreement with what was predicted in the guidelines.

### 1.2. Pattern-Based and Point Based Systems

Two of the eight RSSs described above, ACR- and C-TIRADS, are point-based systems and the six others are pattern-based. Of note, however, another point-based system was published in 2011, sometimes referred as “Kwak-TIRADS”, and has gained acceptance in some parts of South Korea, in China, and other countries or regions [22]. The TI-RADS scores ranged from 1 to 5, with 1 corresponding to no nodule, 2 and 3 to benign and probably benign with no suspicious US features, and then 4a, 4b, 4c, and 5 to 1, 2, 3, or 4 and 5 suspicious US features, respectively. In a retrospective study on 1000 patients [23], a significant association was found between the TI-RADS score and Bethesda classification (*p* < 0.001). Most individuals with TI-RADS 2 or 3 had Bethesda 2 result (95.5% and 92.5%, respectively). Among those classified as TI-RADS 4C and 5, most presented Bethesda 6 (68.2% and 91.3%, respectively; *p* < 0.001). The proportion of malignancies among TI-RADS 2 was 0.8%, and TI-RADS 3 was 1.7%. Among those classified as TI-RADS 4A, proportion of malignancies was 16.0%, 43.2% in 4B, 72.7% in 4C, and 91.3% among TI-RADS 5 (*p* < 0.001), showing clear association between TI-RADS and FNA results.

Pattern-based scoring consists of recognizing a grouping of US features. It is the basis of most RSSs. Pattern-based systems have the advantage of quickness and pedagogy, in the way that they easily show and transmit patterns which are frequently encountered in daily practice. For instance, the pattern of an EU-TIRADS 3 [12] is a nodule with oval shape, regular margins, and isoechoic solid component. It describes common aspects of thyroid nodules and simplifies reality as it groups various patterns into a single recognizable one. However, here also lies its disadvantage as it may sometimes go too far in simplifying. For example, a nodule with taller-than-wide shape is considered as high risk by the EU-TIRADS, regardless of its echogenicity and composition, although its malignancy risk would rather be intermediate. The K-TIRADS tries to overcome this problem by dividing the intermediate category into two, depending on echogenicity and composition [10].

Point-based scoring systems consist of summing points that have been formerly attributed to US features. It is the core of the ACR-TIRADS and of the C-TIRADS. The advantages are that all existing US features can be included and that the system can easily be modified with experience and virtually tested. A disadvantage is the necessity of learning by heart the number of points of each feature and having to sum them for every nodule, which can be quite time consuming if these are numerous and or if the workload is very intense. Another disadvantage is that the point assignment to each US feature is basically arbitrary. Interestingly, the ACR-TIRADS has been the attempt of a revision using artificial intelligence (AI) [24]. A genetic AI algorithm was applied to a training set of 1325 nodules and to create an optimized scoring system. This AI TI-RADS assigned new point values for eight features, including a simplified scheme for some categories. For example, only assigning points to solid nodules and eliminating point assignments to other composition features represented one such modification.

Direct implementation of the calculation algorithm in US machines could significantly simplify the use of both point-based and pattern-based RSSs.

### 1.3. Other Similarities and Differences

The aims of RSSs are identical: provide the highest possible diagnostic accuracy and reduce the number of unnecessary FNAs. All RSSs stratify the risk of malignancy with a qualitative approach ranging from normal to high risk linked to quantitative risk ranges appreciated by clinical studies. However, they differ by the number of classes used, the features defined as highly suspicious and the use of composition and ETE for risk stratification (Table 1).

#### 1.3.1. Lexicon

Lexicons have many similarities, in particular regarding the categories (composition, echogenicity, shape, margins) and terms that have been chosen to describe nodules. In particular, the taller-than-wide shape has a common definition. However, significant differences exist:

Echogenicity: the EU-TIRADS considers that even a small hypoechoic part is sufficient to classify the nodule as hypoechoic, whereas K-TIRADS and ACR-TIRADS define the echogenicity of the nodule by its predominant one in heterogeneous nodules. The correctness of the definition of the K-TIRADSs seems to have been confirmed in a report on 2255 nodules, with a retrospective design [25]. Finally, the term markedly hypoechoic applies in the K-TIRADS as more or of equivalent hypoechogenicity to the strap muscles and in the other systems only as more hypoechoic than strap muscles.

Composition: this not taken into account in the EU-TIRADS. The 2016 Korean Society of Thyroid Radiology–Korean Thyroid Association guidelines recommend the use of solid composition for nodules with no obvious cystic change even considering that nodules with minimal cystic changes (<10%) do not have a high malignancy risk. By the ACR-TIRADS, cystic changes are considered as significant if they represent at least 50% in volume.

Hyperechoic foci: the EU- and K-TIRADS differentiate the ones which are located in the cystic part of the nodule (with a comet-tail artifact in the K-TIRADS), in favor of benignity, opposite to the one in the solid part, whereas the ACR-TIRADS does not discriminate between the two.

#### 1.3.2. Classification

The lowest grade has different definitions from one RSS to another: in K- and EU-TIRADS, class 1 corresponds to a normal examination whereas it means benign for the ACR-TIRADS, very low suspicion for the ATA, and low risk for the AACE system. The number of classes varies from 3 in the AACE to 4 for the ATA and 5 for most other systems.

#### 1.3.3. Patterns

Spongiform and purely cystic nodules are universally recognized as benign or at very low risk. Microcalcifications, taller-than-wide shape, marked hypoechogenicity, and irregular margins are also widely considered as weighing a high risk of malignancy. However, the K-TIRADS and the ATA system consider that these features as high risk only in solid hypoechoic nodules, whereas they are considered to be so in all nodules for the EU-TIRADS and AACE/ACE/AME system. In the ACR-TIRADS, a taller-than-wide shape, very hypoechoic and punctate echogenic foci are attributed 3 points but irregular margins only 2. Risk stratification differences are illustrated on Figure 1.

### 1.4. Raw Diagnostic Values in Comparative Studies (before Applying Size Cut-Offs for the Decision to Perform FNA)

Many studies have attempted to compare the systems with each other. In particular, a comparison was performed between the BTA, AACE, and ATA RSSs [26]. The conclusions were that classification systems had elevated positive predictive value of malignancy in high-risk classes. ATA and AACE/ACE/AME systems were effective for ruling-out indication to FNA in low US risk nodules. A similar diagnostic accuracy and a substantial inter-observer agreement was provided by the 3- and the 5-category classifications.

Systematic reviews and meta-analyses are available. In a study including 10,437 thyroid nodules and 12 studies on different TIRADS, a pooled sensitivity of 0.79 and a pooled specificity of 0.71 were found [27]. Subgroup analyses showed that the most important factor of heterogeneity in studies was the final diagnostic references (histological and cytological standards or only histological results). In the report by Kim et al. [28], a total of 29 articles including 33,748 thyroid nodules met the eligibility criteria and were included in the analysis. The report concluded that the overall diagnostic performance of the four US-based risk stratification systems (ACR, ATA, K, and EU-TIRADS) was comparable. However, most of these studies used cytology as a gold standard and eliminated indeterminate ones of the assessment, thus introducing a significant recruitment bias. An interesting report used histology as a gold standard while comparing the ACR- and EU-TIRADS [29]. It was found that ACR-TIRADS and EU-TIRADS score had similar and satisfactory accuracy values for predicting thyroid malignancy (AUC: 0.835 for ACR-TIRADS vs. 0.827 for EU-TIRADS).

Thus, up to now, no RSS has shown a real significant advantage over the others in terms of raw diagnostic value.

### 1.5. Inter-Observer Agreement

One of the main aims of the TIRADS was to improve interobserver discrepancies in the description of US features. Yime et al. reported that the concordance rate of nodules classified as high- or intermediate-suspicion was high (84.1–100%), but low or mildly-suspicious nodules exhibited relatively low concordance (63.8–83.8%) between the K-TIRADS, ATA, and ACR-TIRADS [30].

In a blinded multicenter study [31], thyroid nodules were classified according to AACE/ACE/AME, EU-TIRADS, ATA, and ACR-TIRADS US classifications. Intra- and interobserver agreement was calculated using cross-tabulation expressed as mean Cohen’s Kappa (K-coefficient). It was judged that intraobserver reproducibility for thyroid nodule US reporting and US classification systems appears fairly adequate, while the interobserver agreement between different centers is lower than in single-center trials. Reporting and rating ability of thyroid US examiners still appeared inconsistent.

The impact of radiologist experience was evaluated for the ACR-TIRADS [32]. Three experienced and three less experienced radiologists assessed 150 thyroid nodules using the TI-RADS lexicon. Concordance was significantly higher for less experienced readers in identifying margins (84.3% vs. 67.4%), echogenic foci (76.9% vs. 69.3%), comet tail artifact (89.6% vs. 79.2%), and punctate echogenic foci (85.3% vs. 75.5%), and lower for peripheral rim calcifications (95.0% vs. 97.8 %), but was not different for the remaining categories and features. However, the overall TI-RADS level and recommendation for FNA were unaffected, supporting the robustness of the TI-RADS lexicon and its continued use in practice.

In a study comparing Kwak, ACR, and EU-TIRADS and ATA system [33], it was found that after a first session and a consensus reading, interobserver agreement (IA) significantly increased but did not affect the diagnostic accuracy. Interobserver agreement and diagnostic accuracy were very similar for the four investigated risk stratification systems.

Finally, in a study comparing Kwak- and EU-TIRADS [34], it was found that the interobserver agreement (Cohen’s κ) was 0.52 and 0.67 for Kwak-TIRADS and EU-TIRADS, respectively, and rated as substantial.

The synthesis of all these reports leads to believe that interobserver agreement is very similar for all systems: fairly adequate to robust. However, it may be better for high- or intermediate-suspicion nodules than for lower mildly-suspicious ones. Moreover, while intraobserver reproducibility for thyroid nodule US reporting and US classification systems appears fairly adequate, the interobserver agreement between different centers may be lower than previously assessed in single-center trials. Thus, and unfortunately, it seems that despite what was expected, reporting and rating ability of thyroid US examiners is not much better for classification systems than it is for individual US features. Dedicated training is necessary and proven to be able to achieve this goal.

## 2. Indications for FNA and Diagnostic Values of RSSs after Applying Size Cut-Offs for FNA

### 2.1. Dimensional Cut-Offs

Each US risk-stratification system has set its own cut-offs for guiding fine needle aspiration cytology indications (Table 2).

For BTA guidelines [3], US appearances that are indicative of a benign nodule (U1–U2) should be regarded as reassuring not requiring FNA, unless the patient has a statistically high risk of malignancy (i.e., age less than 20 or older than 60 years; firmness of the nodule on palpation; rapid growth; fixation to adjacent structures; vocal cord paralysis; regional lymphadenopathy; history of neck irradiation; family history of thyroid cancer). US guided FNA is indicated for all U3–5 nodules (i.e., equivocal, indeterminate or suspicious of malignancy nodules), independent of their size. Cytologically benign nodules with indeterminate or suspicious US features should undergo repeat FNA for confirmation, due to the significant rate of malignancy. Nodules with FDG uptake should be investigated with FNA unless the patient has limited life-expectancy.

AACE guidelines [6] recommend FNA for high US risk thyroid lesions ≥10 mm and intermediate US risk thyroid lesions >20 mm. For low US risk thyroid lesions, FNA is recommended only when size is >20 mm and increasing or associated with a risk history and before thyroid surgery or minimally invasive ablation therapy. These guidelines highlight that nodules <5 mm should be monitored with US, rather than biopsied, irrespective of their sonographic appearance, in light of the low clinical risk. For nodules measuring 5–10 mm, FNA sampling or watchful waiting can either be considered according to the clinical setting and patient preference. US-guided FNA is recommended for subcapsular or paratracheal nodules, suspicious lymph nodes or suspicion of extrathyroidal spread, positive personal or family history of thyroid cancer, or coexistent suspicious clinical findings (e.g., dysphonia). A retrospective series on 859 FNA from 598 patients showed that moderate-to-elevated risk of malignancy (i.e., Bethesda III to VI categories) [35] lesions would have been missed for 13 out of 17 nodules, if intermediate risk nodules <20 mm had been excluded from FNAC, of which 11 were malignant at definitive histology [7]. Cytological confirmation of diagnosis would have been missed in 8 out of 26 cases of high-risk lesion <10 mm if a watchful waiting attitude has been chose over FNA sampling.

ATA guidelines [8] recommend FNA for nodules ≥10 mm in greatest dimension with high and intermediate suspicion US pattern, ≥15 mm in case of low suspicion US pattern, and ≥20 mm in greatest dimension with very low suspicion US pattern (e.g., spongiform). Alternatively, observation without FNAC is also stated as a reasonable option. FNA is not required for purely cystic nodules.

The 2016 revised Korean Society of Thyroid Radiology Consensus Statement and Recommendations [10] stated that FNA should be restricted to K-TIRADS 2 spongiform nodules ≥ 20 mm, K-TIRADS 3 ≥ 15 mm, K-TIRADS 4 or 5 ≥ 10 mm, and in selective cases of K-TIRADS 5 > 5 mm. Applying cut-off of ≥10 mm for K-TIRADS categories 4 or 5 and ≥15 mm for K-TIRADS 3, the negative predictive value was 94.3% according to a study where 85.5% of the malignant tumors were papillary thyroid cancer (PTC) [36], meaning less than 6% of missed carcinomas.

Because the false negative rate of an initial benign findings of FNA could be relatively high (11.3–56.6%) for thyroid nodules with suspicious US features [10], FNA should be repeated in these cases within 6–12 months after the initial FNAC.

The European Thyroid Association Guidelines [12] stated that FNA should usually be performed only for nodules EU-TIRADS 3 > 20 mm, EU-TIRADS 4 > 15 mm, and Eu-TIRADS 5 > 10 mm. Patients with highly suspicious EU-TIRADS 5 nodule < 10 mm can have the choice between active surveillance or immediate FNAC if surgery is decided.

Regarding EU-TIRADS 3 nodule, it should be pointed out that entirely solid isoechoic nodules can correspond to follicular cancer or a follicular variant of PTC [36] in <4% of cases. As a consequence, few carcinomas will be missed after applying FNA cut-off for EU-TIRADS 3 nodules.

The ACR committee [16] recommends FNA for TI-RADS 5 nodules (7 points or more) ≥ 10 mm, for TI-RADS 4 nodules (4 to 6 points) ≥ 15 mm, for TI-RADS 3 nodules (3 points) ≥ 25 mm. FNA is not indicated for TI-RADS 1 (0 point) and TI-RADS 2 (2 points) nodules regardless of their size. The 10 mm size-threshold to indicate FNA for highly suspicious nodules is consistent with most other guidelines. However the ACR thresholds for mildly and moderately suspicious nodules (25 mm and 15 mm respectively) are higher than the cut-offs advocated by the ATA and the Korean Society of Thyroid Radiology. Rational for the ACR cut-offs relies on the one hand, on the discrepancy regarding the size of PTC at definitive histology (26.5 ± 10.7 mm) and the size on ultrasound (19.7 ± 11.7 mm) on a retrospective series including 205 PTC [36]. However, all series reporting outcome of thyroid cancers are based on the size of resected specimen. On the other hand the ACR cut-offs rely on a slight decrease in 10-year thyroid cancer-specific survival for nodules ≥30 mm [37].

The Chinese Guidelines for US Malignancy Risk Stratification of Thyroid [20] does not recommend FNA for TIRADS 2 and 3 nodules but recommend FNA for TIRADS 4A nodules > 15 mm. FNA is recommended for TIRADS 4B or 4C or 5 nodules > 10 mm. In case of parameters predictors of poor prognosis of PTC such as multifocality, or nodule(s) immediately adjacent to the trachea or recurrent laryngeal nerve, then US-guided FNA can be considered if the nodule is TIRADS 4A > 10 mm or TIRADS 4B or 4C or 5 > 5 mm. If TIRADS 4B or 4C nodules <5 mm are multiple, or are immediately adjacent to the capsule, trachea, or the recurrent laryngeal nerve, then biopsy is required by comprehensively considering the skills of the doctor and the anxiety level of the patient. For patients with familial thyroid carcinoma or history of radiation exposure during childhood, the size threshold for FNA can be appropriately reduced. These guidelines recommend taking into account the patient’s personal preference and anxiety level to determine if FNA is appropriate in each specific case.

Apart from the BTA classification system, which had not defined size cut-offs for the selection of nodules that should be submitted to FNA, most US RSS agree on the threshold of 10 mm for indicating FNA in highly suspicious nodule, although some RSS (AACE, KThRS) recommend FNA for nodules between 5 and 10 mm in selected cases and systematically for Chinese TIRADS. The threshold for indicating FNA for intermediate-risk nodule varies from 10 mm for the ATA classification system and the K-TIRADS, to 15 mm for the ACR classification system and the EU-TIRADS and even 20 mm for the AACE classification system. A low threshold of 5 mm for intermediate risk nodules is only recommended by the Chinese TIRADS. The threshold for indicating FNA for low suspicion nodule varies from 15 mm for ATA classification system, K-TIRADS and Chinese TIRADS, to 20 mm for EU-TIRADS and even 25 mm for ACR TIRADS. Most RSS do not recommend FNAC for very low suspicion nodules, i.e., AACE (FNAC can be performed in selective cases), ATA, ACR classification systems, EU-TIRADS, and Chinese TIRADS. Only K-TIRADS and ATA guidelines retain FNA indication for spongiform or partially cystic nodules ≥ 20 mm.

### 2.2. Diagnostic Value after Applying Cut-Offs: Decision Guidance, Avoided FNAs, and Missed Carcinomas

Different retrospective series demonstrated that a very small proportion of thyroid cancers are missed after applying size cut-offs for FNA of the ACR committee. This would decrease with lower cut-offs, but create a substantial increase in the number of benign nodules that would be explored. A series showed that 13 cancers (11 PTC, one follicular and one medullary thyroid cancer) among nodules measuring 15–25 mm, would have been missed if FNA would have not been performed for the 874 nodules measuring less than 25 mm included in this series [38]. Middleton et al. showed, in a series of 3822 nodules Bethesda II or VI, that among 352 malignant nodules (303 were histologically confirmed), 40 nodules would have received a recommendation for no further evaluation, among which were 16 malignant nodules ≥10 mm [38].

A recent meta-analysis [39], including 12 studies [7,13,26,40,41,42,43,44,45,46,47,48] representing 18,750 nodules, evaluated the ability of 5 US RSSs (AACE, ACR, ATA, EU-TIRADS, K-TIRADS) for the appropriate selection of thyroid nodules for FNA. Diagnostic odds ratio, representing the test performance and corresponding to the odds of the FNA being indicated in a malignant nodule compared to the odds of the FNA being indicated in a benign one, were calculated for each US RSS. Keeping in mind that data on AACE and EU-TIRADS were sparse, diagnostic odd ratio was higher for ACR-TIRADS in comparison with the other systems. The higher discriminative power was related to a higher ability of ACR-TIRADS to select malignant nodules for FNA, while no difference was found for benign nodules. This cannot be explained by the size cut-offs for FNA in intermediate- and high-risk-nodules, given that it is similar to that of the other US RSSs. However, fewer nodules will probably be classified as intermediate- or high suspicious than in other systems, because of the point-based pattern of this RSS. As intermediate risk nodules are frequent, this could explain the advantage of the ACR-TIRADS over the other systems.

For example, in the series of Xu et al. [43], comparing the diagnostic value of three RSS (i.e., ACR-, EU- and K-TIRADS) in 2465 thyroid nodules, the rate of unnecessary FNA was lowest with the ACR-TIRADS (17.3%), followed by ETA-TIRADS (25.2%), and K-TIRADS (32.1%). Among nodules not submitted to FNA, 33.1%, 37.7%, and 38.2% thyroid cancers would be missed by the same TI-RADS, respectively. Finally, after applying adequate FNA cut-offs of each of these TI-RADS, 62.6%, 54.6%, and 43.9% FNAC were avoided, respectively.

In the work by Grani et al. [44] that prospectively compared the performances of five internationally endorsed sonographic classification systems (those of the ATA, the AACE, the ACR, the ETA, and the KSThR) in 477 patients, application of the systems’ FNA criteria would have reduced the number of biopsies performed by 17.1% to 53.4% (17.1% for K-TIRADS, 30.7% for EU-TIRADS, 34.9% for AACE, 43.8% for ATA, and 53.4% for ACR TIRADS). The percentage of missed carcinomas was low comprised between 2.2% for ACR TIRADS and 4.1% for ATA.

In the work of Yoon et al. [49] comparing the diagnostic performance of US-guided FNAC criteria for detecting malignant thyroid nodules in ACR TI-RADS and EU-TIRADS, the percentage of unnecessary FNAC was estimated at 53% for the EU-TIRADS and 28% for the ACR-TIRADS.

As a conclusion, all RSSs seem to reduce effectively the number of unnecessary FNAs. However, this is at the cost of temporarily missing a significant proportion of carcinomas. Their diagnosis will be postponed until they eventually grow and are then diagnosed after they reach the cut-off threshold defined for FNA according to their US risk category. Most of the time, this strategy implies no significant loss of chance for the patient. This is due to the statistical predominance of papillary carcinomas of low and intermediate risks among all thyroid cancers. However, looking for lymph node or extra-thyroidal extension, including clinical factors such as age, sex, personal and family history with risk factors of thyroid cancer, tumor growth rate, and also serum calcitonin whenever judged relevant is critical for making the right decision to prevent missing more aggressive carcinomas. Thus, the recommendation for no further evaluation, as specifically formulated in the ACR-TIRADS, should be considered with caution and put into perspective including clinical and biological data.

## 3. Weaknesses of TIRADSs

### 3.1. Insufficient Sensitivity for the Diagnosis of Follicular Thyroid Carcinoma and Follicular Variant of PTC

While historically the follicular variant of PTC (FVPTC) was considered a diagnostic pitfall of US, this notion was not confirmed in a report published in 2018 on 34 cases [50]. The K-TIRADS score was 3, 4, and 5 in 5.9%, 2.9%, and 91.2%, respectively. Thus, the false negative rate does not seem to exceed 6%.

In a study on 45 follicular thyroid carcinomas (FTCs) from 45 consecutive patients, with a median tumor diameter of 32 mm, an ovoid isoechoic nodule with or without lobulated margins was the most frequent presentation [51]. When FTCs were classified according to RSSs, the most common categories were intermediate and high risk, though 1 out of 3 cases was not classifiable. FTCs were classified as high risk/high suspicion/malignant in 11% to 74% of cases, with a statistically significant difference among the systems. More specifically, 26.7% were classified as EU-TIRADS 3 but all submitted to FNA due to their size and 2.2% and 26.7% were classified as ACR-TIRADS 2 and 3, respectively and among these 25% were not submitted to FNA, also due to size cut-offs. To conclude, in FTCs cases, the RSSs false negative rate seems persistently higher than for FVPTCs, around 25%. Clinicians should be aware of this, especially in the era of thermal ablation, to try to avoid treating such nodules by alternatives to surgery. More specifically, exclusively solid isoechoic and mildly hypoechoic nodules should always be considered with caution.

### 3.2. Insufficient Specificity to Rule-Out Autonomously Functioning/Hot Thyroid Nodules from FNA

Autonomously functioning thyroid nodules (AFTN) account for 5–10% of palpable lesions and are very rarely malignant. In a study on 87 AFTNs from 85 consecutive patients who had undergone US, scintigraphy, and thyroid function evaluation, AFTNs were reclassified according to AACE/ACE/AME, ACR-TIRADS, ATA, BTA, EU-TIRADS, K-TIRADS, and TIRADS [52]. An ovoid isoechoic nodule with median diameter of 22 mm (range 10–59) was the most frequent US presentation. When AFTNs were reclassified according to US RSSs, the most common categories were low and intermediate risk. AFTNs were assessed as being at high risk/high suspicion/malignant in 1–9%, with good agreement among AACE/ACE/AME, ATA, EU-TIRADS, K-TIRADS, and TIRADS. Remarkably, FNA was indicated in 27–90% of AFTNs. It was concluded that ultrasound RSSs prompt inappropriate FNA in a significant number of patients with AFTN. The management strategy of thyroid nodules being essentially based on US risk stratification and size cut-offs, it could be considered that, depending on the RSS used, 2.7% to 9% of all nodules should have been excluded from FNA.

However, the reverse strategy of submitting all TNs to scintigraphy to exclude an AFTN before US exploration would drastically augment the costs with no diagnostic gain in, at least, 90% of all nodules.

### 3.3. High Rates of Nodules Classified at Intermediate Risk (Usually TI-RADS 4)

Based on the high negative predictive value of all RSSs, it could be considered that FNA could be avoided for most nodules classified as low risk, especially for those of mixed composition. At the opposite end, the high positive predictive value of high-risk categories prompt the indication for FNA in most cases if the size is over 10 mm, knowing these represent a minority of all nodules.

Conversely, the indication for FNA in intermediate risk nodules is still a matter of concern. Indeed, these nodules represent a substantial part of all nodules discovered during US thyroid imaging and even a more substantial part of those referred for FNA. Using the ATA US pattern risk assessment, nodules were classified as intermediate risk in 31% of cases [9]. Regarding the AACE, 56.9% were considered at intermediate risk in another report [7]. In a study on 305 nodules with final histology as gold standard, it was shown that ACR-TIRADS 4 nodules represented 28.8% of all nodules and EU-TIRADS 4 category 22% [29]. Finally, in a study with a prospective design with cytological examination as a gold standard on 4550 nodules [53], the rate of TIRADS 4A nodules (equivalent to EU-TIRADS 4) was 44.5%.

Thus, the main difficulty in significantly and appropriately reducing the indications for FNA is the high rate of intermediate risk nodules. Research has been performed to improve the low specificity of the category for the diagnostic of malignancy by using either Doppler or elastography. In a report on 80 nodules, no significant differences were observed in elasticity score or strain ratio between benign and malignant nodules [54]. 18F-FDG PET/CT could be a more useful tool to discriminate intermediate risk nodules [55]. 18F-FDG PET/CT showed 85.7% sensitivity and 41.4% specificity. Thus, 18F-FDG PET/CT may have a role in stratifying the cancer risk of thyroid nodules with an intermediate ultrasound assessment. More specifically, thyroid lesions classified as EU-TIRADS 4 without 18F-FDG uptake could be ruled out from further examination. Further prospective and cost-effectiveness studies are however needed.

### 3.4. Thyroid Diffuse Masses

All RSSs have been studied and developed for nodules. However, it is unclear whether diffuse thyroid masses have been taken into account in those systems. These are most of the time responsible for pressure symptoms with a rapid development. The most common US presentation is a hypoechoic mass invading one lobe or all the thyroid gland. It is usually hypoechoic, with poorly defined margins. Vascularity and stiffness are variable and they can be accompanied or not by suspect cervical lymph nodes. The following main aspects should be considered:First, several etiological hypotheses should always be mentioned in the US report, including anaplastic carcinoma, lymphoma, metastases from non-thyroidal origin, and large differentiated papillary and follicular carcinomas. Riedel’s thyroiditis could be added to this list. In this case, marked hypoechogenicity and absorption of the US beam, absence of vascularity, and high stiffness are relatively characteristic features.The context helps refining the hypotheses. Knowledge of a prior renal cell carcinoma is for instance in favor of a metastasis and rapid development in an elderly subject with severe pressure symptoms in favor of an anaplastic carcinoma.Core-needle biopsy or surgical biopsy, depending on the center’s habits, should systematically be added to FNA, due to its low diagnostic power in this situation.Quick referral to a tertiary care center is advised.

### 3.5. Absence of Validation in Large Non-Specialized Medical Communities

One of the main issues in adopting RSSs in daily life is the limited evidence regarding their diagnostic value when applied by non-specialized teams, most of the available literature on the subject being produced by expert centers. Studies carried out outside the specialized world of thyroid imaging without dedicated US machines are necessary to confirm the real world efficiency of all RSSs.

## 4. Pitfalls

### 4.1. Shrinking Nodules

Nodules with a cystic or hemorrhagic component can evolve by shrinking. Risk factors for such evolution include abundant blood supply, non-smooth margin of the internal solid portion, and a spongiform internal content [56]. The process can be of variable length, sometimes lasting for years, but frequently leads to ambiguous US features mimicking malignancy. Such nodules often harbor a taller-than-wide shape, marked hypoechogenicity or some hyperechoic spots and can easily be classified at high risk of malignancy, whatever the RSS used. Some sonographic imaging features, such as regular eggshell calcifications, peripheral hypoechoic or hypoechoic rim, posterior shadowing, and absence of intranodular vascularization have been described [57] to help diagnosing this pattern, named “mummified thyroid syndrome” and later on “degenerating thyroid nodules” [58]. Knowledge, if available, of previous images showing the thyroid nodule shrinkage over time is useful for reaching the correct final diagnosis. In case of doubt, FNA of such suspicious thyroid nodules and sonographic follow-up contribute to establishing the final diagnosis of benign thyroid findings. The cytology is mainly composed of thick colloid and macrophages and the cytopathologist should be informed of the hypothesis. Otherwise, the result could be considered as non-diagnostic instead of representative of the lesion [59].

### 4.2. Subacute Thyroiditis

Subacute thyroiditis can also mimic malignancy by US, because frequently displaying a taller-than-wide shape and marked hypoechogenicity. However, the existence of spontaneous thyroid pain, low TSH, and elevated serum inflammatory markers frequently allows the diagnosis. On the US point of view, it has been shown that the lesions have poorly defined margins that can help differentiating from a carcinoma [60]. In case of persistent doubt, it is advised to proceed to FNA if TSH is normal, or to scintigraphy if TSH is low, which will show an absence of tracer uptake. US follow-up is also advised, showing progressive regression of the hypoechoic zone and absence of a true nodule that could also have been hidden initially by the marked hypoechogenicity of the lesions.

### 4.3. Confusion or Absence of Clear Distinction between Nodular Disease and Hyperplasia

Hyperplasia of the follicular epithelium is the most common morphological change in the thyroid seen by the pathologist [61]. The manifestation of this process is the goiter (diffuse or nodular hyperplasia). The US features range from a simple isoechoic enlargement of the thyroid gland to multiple coalescent isoechoic nodules, usually of small size individually with no or poor definite margins. This pattern is very frequent in regions of endemic goiter. Solely did the EU-TIRADS address this issue, but it should be included in the future in RSSs, because of its very low risk of malignancy and of the feeble interest of FNA, that may even lead to false positive results [62].

## 5. Suggestions for the Future

### 5.1. Absence of Classification for TNs Treated with Thermal Ablation

Thermal ablation, especially laser and radiofrequency (RFA), is of increasing use in the treatment of benign thyroid nodules and is considered as a possible alternative to surgery [63]. In a systematic review, it has been shown that RFA induces a volume reduction ratio ranging between 66.9% and 97.9% three years after the procedure [59]. These treatments induce important changes in the US features of nodules that can mimic malignancy. Nodules turn solid and hypoechoic, even markedly hypoechoic, sometimes with irregular margins and calcifications [64]. As radiofrequency is of frequent use for liver tumors, the LI-RADS Treatment Response (LR-TR) algorithm was introduced in 2017 to assist radiologists in assessing hepatocellular carcinoma (HCC) response following locoregional therapy [65]. A comparable addendum should be part of future thyroid RSSs.

### 5.2. Incorporating in the Algorithm the Number of Nodules Especially If They Belong to the Same Category

Different studies demonstrated that a single nodule increases the risk of malignancy compared to multiple nodules [29,66]. Moreover, this parameter has high inter-observer agreement and is easy to implement. Taking into account in the algorithm of future US RSSs the number of nodules to decrease the estimated risk of malignancy, especially if all are low to intermediate risk nodules, could be valuable.

### 5.3. Taking into Account Age, Sex, Time Since Discovery, Results of Previous FNAs

Many risk factors for thyroid nodules malignancy have been suggested, such as patient age, sex, nodule size, and composition, but our understanding of the specific risk attributable to these is not precisely known. An interesting study [66] demonstrated in 20 001 thyroid nodules evaluated by FNA from 1995 to 2017 a significant increased risk of malignancy for patient age >52, male sex, nodule size with growing risk from 20 mm until more than 40 mm in comparison with nodules less than 20 mm. On the opposite side, cystic content (at least 25% of the nodule) was associated with a decreased risk of malignancy compared with predominantly solid nodule, as well as the presence of additional nodules with lowest risk for greater than 4 nodules. Interestingly, a free online calculator was constructed to provide malignancy-risk estimates based on these variables.

### 5.4. Taking into Account the Serum Value of TSH (to Exclude a AFTN) and Calcitonin (to Detect a Medullary Cancer), When Available

Serum TSH should be measured during the initial evaluation of a patient with one or more thyroid nodule(s). If the serum TSH is low, a radionuclide (preferably 123I) thyroid scan should be performed to exclude AFTN from FNAC and to explore the etiology of hyperthyroidism, provided that there is no evidence of Graves’ disease. In case of normal serum TSH value, there are no US features correlated with autonomous nodules [67,68]. The cost-effectiveness of submitting all nodules to a thyroid scan to avoid unnecessary FNA for AFTN is questioned.

Calcitonin may detect C-cell hyperplasia and medullary thyroid cancer (MTC). However, most guidelines cannot recommend for or against routine calcitonin measurement in patients with thyroid nodules. A recent review [69] demonstrated that calcitonin has good sensitivity and specificity to diagnose MTC and could be useful when available in the evaluation of thyroid nodules. The literature and the experience show that for a calcitonin level over 100 pg/mL nodule larger than 1 cm are MTC. For levels below 100 ng/L and that in nodules larger than 1 cm the systematic calcitonin measurement does not bring a clear advantage for the diagnosis, especially if at low or intermediate US risk. However, the value of routine testing in patients with thyroid nodules remains questionable, due to the low prevalence of MTC, and whether routine calcitonin testing improves prognosis in MTC patients remains unclear. In clinical practice, situations associated with false positivity of calcitonin tests (e.g., renal insufficiency, treatment with proton pump inhibitor, obesity) and the correlation of calcitonin value with the nodule volume should be taken into account for the interpretation of the result. Calcitonin measurement remains mandatory in case active surveillance of EU-TIRADS 5 nodules or proven microcarcinomas is considered and before surgery or thermal ablation. Regardless, the heterogeneous US presentation of MTC [70] and the low sensitivity of FNA in detecting MTC [71] has to be taken into account during the clinical practice.

### 5.5. D Vascularity

Advanced ultrasound techniques may improve the risk estimation and could be used more extensively. For example, Borlea et al. [72] demonstrated that adding 4D vascularity to the French TIRADS score proved beneficial for predicting the malignancy risk and may add important knowledge in uncertain situations.

An international team has been set up and is currently working on a global new TIRADS, to be called I-TIRADS for International TIRADS. It will include a lexicon, an RSS, and recommendations for FNA and follow-up. Maybe some of these suggestions could be taken into account to create this new version. The pitfalls they imply are detailed in Table 3.

## 6. Conclusions

The different US RSSs introduced since the late 2000s have facilitated the effective interpretation and communication of thyroid US findings among physicians and cytopathologists and with the patient. On the whole, there are similarities among the different RSS regarding the lexicons used and the categorization of nodules, although differences and specificities remain. Diagnostic performance and efficacy of FNA performed according to the different RSS vary, mainly influenced by different size cut-offs and partially by different risk categorizations of nodules. Understanding the strengths and weakness of the different RSSs will help to improve each system and may provide the basis for an ultimate international standardization. Efforts should be made to merge the different systems utilized around the world with the ultimate aim of eliminating unnecessary thyroid biopsies without jeopardizing the detection of clinically significant malignancies.

## Figures and Tables

**Figure 1 cancers-13-04316-f001:**
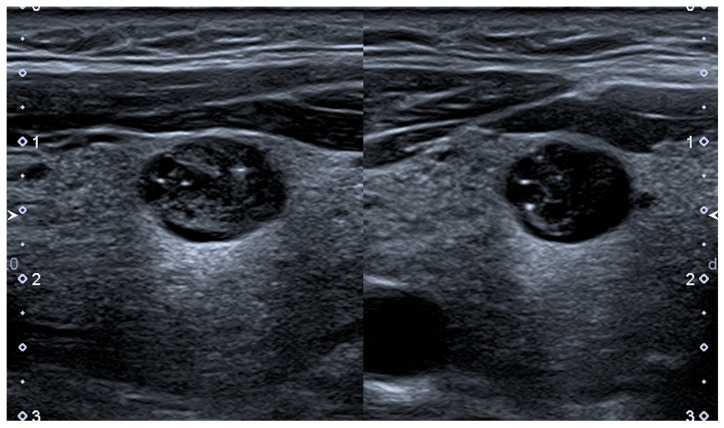
Longitudinal (**left** picture) and transverse (**right** picture) of an oval shaped, isoechoic nodule of mixed composition with hyperechoic spots located at the bottom of microcystic cavities. The nodule measures 11 × 9 × 7 mm. Classification: Chilean-TIRADS 2, Kwak-TIRADS 4a, BTA U3, AACE Class 2, ATA not classifiable as being of mixed composition with hyperechoic spots, K-TIRADS 4, EU-TIRADS 3, ACR-TIRADS 4, C-TIRADS 4A. Note: magnification × 3; scale bar: 1 cm/unit; TIRADS = Thyroid Imaging and Reporting Data System.

**Table 1 cancers-13-04316-t001:** Comparison of some specificities of existent risk stratification systems (RSSs). Note: ETE = extra-thyroidal extension; RSS = risk stratification system.

RSS	Number of Classes	Meaning of TIRADS 1	Pattern or Point-Based RSS	Features of High Suspicion	Composition Included in the RSS	ETE Included in the RSS
Chilean TIRADS	6 TIRADS 4 divided into 2 subclasses	Normal examination	Pattern	Irregular margins Irregular shape Multiple peripheral microcalcifications Penetrating vessels	Yes	No
Kwak-TIRADS	5 TIRADS 4 divided into 3 subclasses	No nodule	Point	Marked hypoechogenicity Irregular margins Microcalcifications Taller than wide	No	No
BTA	5	Normal	Pattern	In a solid hypoechoic nodule: Irregular margins Microcalcifications Globular calcifications Intranodular vascularity Taller than wide Lymphadenopathy	Yes	No
AACE	3	Low risk	Pattern	Marked hypoechogenicity Irregular margins Microcalcifications Taller-than-wide Extrathyroidal growth Pathologic lymph node.	No	Yes
ATA	5	Benign	Pattern	In a solid hypoechoic nodule: Irregular margins Microcalcifications Taller than wide Rim calcifications with small extrusive soft tissue component Extra-thyroidal extension	Yes	Yes
K-TIRADS	5	Absence of nodule	Pattern	In a solid hypoechoic nodule: Irregular margins Microcalcification Nonparallel orientation	Yes	No
EU-TIRADS	5	Absence of significant nodule	Pattern	Marked hypoechogenicity Irregular margins Microcalcifications Taller than wide	No	No
ACR-TIRADS	5	Benign	Point	Marked hypoechogenicity All punctate echogenic foci Taller-than-wide Extra-thyroidal extension	Yes	Yes
C-TIRADS	5 TIRADS 4 divided into 3 subclasses	No nodule	Point	Markedly hypoechogenicity Ill-defined and irregular margins Vertical orientation Solid composition Microcalcifications Extra-thyroidal extension	Yes	Yes

For abbreviations of the names of the RSSs, please refer to Section 1.1. Note: ETE = extra-thyroidal extension; RSS = risk stratification system.

**Table 2 cancers-13-04316-t002:** Size cut-offs for performing FNA recommended by each RSS.

RSS	TIRADS 2 or Very Low Risk	TIRADS 3 or Low Risk	TIRADS 4 or Intermediate Risk	TIRADS 5 or High Risk	Small Nodules < 10 mm
Chilean TIRADS	No FNA or follow-up	FNA (no cut-off) or follow-up	FNA (no cut-off)	FNA (no cut-off)	FNA if >3–4 mm and feasible
Kwak-TIRADS	No FNA	No FNA	TIRADS 4a: ≥25 mm TIRADS 4B: 15 mm	TIRADS 4 C and 5: ≥10 mm	No FNA
BTA	No FNA	All nodules	All nodules	All nodules	-
AACE	No FNA	≥20 mm and growing lesion or risk factors	≥20 mm	≥10 mm	<5 mm no FNA 5–10 mm FNA if clinical or US risk factors or PP
ATA	≥20 mm or observation	≥15 mm	≥10 mm	≥10 mm	5–10 mm FNA if clinical or US risk factors or PP
K-TIRADS	≥20 mm	≥15 mm	≥10 mm	≥10 mm	≥5 mm selective cases
EU-TIRADS	No FNA	>20 mm	>15 mm	>10 mm	FNA or active surveillance, PP
ACR-TIRADS	No FNA	≥25 mm	≥15 mm	≥10 mm	No FNA
C-TIRADS	No FNA	No FNA	≥15 mm	≥10 mm	US risk factors

Note: PP = patient’s preference.

**Table 3 cancers-13-04316-t003:** Current pitfalls of most known risk stratification systems (RSSs) of thyroid nodules and recommendations to improve these.

Current Pitfalls of Existent RSSs Variables Not Taken into Account for Risk Stratification	Suggested Correction
Modifications of nodules treated by thermal ablation are classified as highly suspect	Incorporate a treatment response (TR) algorithm
The number of nodules is an independent predictor of the malignancy risk	Add the number of nodules in the risk stratification algorithm, especially if they look all alike and are of low or intermediate risk
Some clinical variables and previous results of FNA(s) are predictors of the malignancy risk	Incorporate age, sex, time since discovery, results of previous FNAs in the risk stratification algorithm
TSH and serum calcitonin are predictors of the malignancy risk	Incorporate TSH and serum calcitonin in the risk stratification algorithm
Complementary tools not used in most RSSs, such as vascularity and elastography	At least, incorporate these in the lexicon, to allow comparative studies on the subject

## Data Availability

Not applicable.

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
