# Peer review of "The New Era of TIRADSs to Stratify the Risk of Malignancy of Thyroid Nodules: Strengths, Weaknesses and Pitfalls"

_cancers, 2021, doi:10.3390/cancers13174316_

Round 1

Reviewer 1 Report

Review: The new era of TIRADSs to stratify the risk of malignancy of 2

thyroid nodules: strengths, weaknesses and pitfalls

Gilles Russ, Pierpaolo Trimboli and Camille Buffet

The work is well organized and a significant contribution to the field of thyroid diagnostics. Therefore, I recommend the paper for publication. However, there a few changes necessary (especially English language style and grammar, please check again):

  1. Line 54f: As described in section 3.2, the functional status of the TN is important to know, therefore and besides laboratory tests, thyroid scintigraphy should be a part of the diagnostic algorithm. I recommend mentioning that in the introduction section.
  2. 1.1 L 73ff: Are you sure that for all 1097 histological results were available? Is it possible that only thyroid nodules with malignant FNA were resected and those with benign cytology were followed-up? What about thyroid nodules with indeterminate FNA?
  3. In section 5, it would be interesting to mention the planned I-TIRADS.
  4. The reviewer recommend to insert another Figure(s) to explain pitfalls; for instance, thyroid nodule after RFA or radioiodine therapy.
  5. 1.2 L 80: Please mind syntax: In 2014, the BTA guidelines for the management of thyroid cancer were introduced.
  6. 1.2 L 97: please delete the comma behind patients
  7. 1.2 L 98f: please use the abbreviation NPV and PPV (both were explained in 1.1.1 L75)
  8. 1.3: In 2016, the AACE ….Thyroid Nodules were released.
  9. 1.4: In 2017, the American Thyroid …….Thyroid Cancer were published.
  10. 1.5 L 139f: ….are scored as K-TIRADS 2 …..classified as K-TIRADS 3
  11. 1.5 L 143: …any suspicious feature…..
  12. 1.5 L 147: please use abbreviation PPV / NPV
  13. 1.6 L 164 and 168: please use abbreviation PPV / NPV
  14. 1.7 L 172: In contrast to most other RSSs…..
  15. 1.8 L 193: please correct in (Realizing that in China….)
  16. 4 L 324 and 326 please check hyphenation
  17. 5 L 361: However, it may be……
  18. 1 L 380: better use independent of their seize (instead of whatever)
  19. 1 L 416f: please correct EU-TIRADS instead of Eu-TIRADS
  20. 1 L 425: …..(TIRADS 2) nodules regardless of their seize
  21. 1 L 459: Most RSSs do not ……..
  22. 2: the first sentence is too long, please rephrase
  23. 2: L 489: please rephrase
  24. 2 L 495: please insert comma behind RADS
  25. 2 L 509 ff: please rephrase
  26. 2 L521: 45 instead of forty-five;
  27. 4 L 582: please delete the second full stop
  28. 1 L 648: please delete REFERENCE
  29. 4 L 695f: please delete the square brackets

Please delete the attached file. It is not completed.

Author Response

Dear Reviewer,
thank you for your interesting and kind comments. Sorry for the grammar imperfections and thank you for taking of your time to handle this. All comments have been addressed according to your advice as follows:

  1. Line 54f: As described in section 3.2, the functional status of the TN is important to know, therefore and besides laboratory tests, thyroid scintigraphy should be a part of the diagnostic algorithm. I recommend mentioning that in the introduction section.
     Sentence was changed to: "Thus, a more thorough algorithm, also including laboratory tests such as TSH and calcitonin and thyroid scintigraphy when deemed adapted, may be sought in the future."

  2. 1.1 L 73ff: Are you sure that for all 1097 histological results were available? Is it possible that only thyroid nodules with malignant FNA were resected and those with benign cytology were followed-up? What about thyroid nodules with indeterminate FNA?
    Unfortunately, the reviewer's comment is perfectly right ! Far from 1097 histological results were available. Sentence was modified accordingly:
    "The TIRADS classification was evaluated in a sample of 1097 nodules (benign: 703; follicular lesions: 238; and carcinoma: 156), among which all nodules with a malignant FNAB result were submitted to surgery, benign ones by FNAB were followed and in the group of patients with indeterminate or follicular lesions, 31% were operated on, and the rest followed."

  3. In section 5, it would be interesting to mention the planned I-TIRADS.
    Thank you for this comment. The following sentence has been added:
    "An international team has been set up and is currently working on a global new TIRADS, to be called I-TIRADS for International TIRADS. It will include a lexicon, a RSS and recommendations for FNA and follow-up. Maybe some of these suggestions could be taken into account to create this new version."

  4. The reviewer recommend to insert another Figure(s) to explain pitfalls; for instance, thyroid nodule after RFA or radioiodine therapy
    The following table was added:

    Current pitfalls of existent RSSs
    Variables not taken into account for risk stratification

    Suggested correction

    Modifications of nodules treated by thermal ablation are classified as highly suspect

    Incorporate a treatment response (TR) algorithm

    The number of nodules is an independent predictor of the malignancy risk

    Add the number of nodules in the risk stratification algorithm, especially if they look all alike and are of low or intermediate risk

    Some clinical variables and previous results of FNA(s) are predictors of the malignancy risk

    Incorporate age, sex, time since discovery, results of previous FNAs in the risk stratification algorithm

    TSH and serum calcitonin are predictors of the malignancy risk

    Incorporate TSH and serum calcitonin in the risk stratification algorithm

    Complementary tools not used in most RSSs, such as vascularity and elastography

    At least, incorporate these in the lexicon, to allow comparative studies on the subject

  5. 1.2 L 80: Please mind syntax: In 2014, the BTA guidelines for the management of thyroid cancer were introduced.
    Thank you and sorry. The sentence was changed accordingly.

  6. 1.2 L 97: please delete the comma behind patients
    Done

  7. 1.2 L 98f: please use the abbreviation NPV and PPV (both were explained in 1.1.1 L75)
    Done

  8. 1.3: In 2016, the AACE ….Thyroid Nodules were released.
    (1.1.3)
    Done

  9. 1.4: In 2017, the American Thyroid …….Thyroid Cancer were published.
    Done

  10. 1.5 L 139f: ….are scored as K-TIRADS 2 …..classified as K-TIRADS 3
    Done

  11. 1.5 L 143: …any suspicious feature…..
    Added (sorry)

  12. 1.5 L 147: please use abbreviation PPV / NPV
    Done

  13. 1.6 L 164 and 168: please use abbreviation PPV / NPV
    Done

  14. 1.7 L 172: In contrast to most other RSSs…..
    Done

  15. 1.8 L 193: please correct in (Realizing that in China….)
    Done (sorry)

  16. 4 L 324 and 326 please check hyphenation
    Done

  17. 5 L 361: However, it may be……
    Done

  18. 1 L 380: better use independent of their seize (instead of whatever)
    Done

  19. 1 L 416f: please correct EU-TIRADS instead of Eu-TIRADS
    Done

  20. 1 L 425: …..(TIRADS 2) nodules regardless of their seize
    Done

  21. 1 L 459: Most RSSs do not ……..
    Done

  22. 2: the first sentence is too long, please rephrase
    The sentence was split into two.

  23. 2: L 489: please rephrase
    The paragraph was rewritten as follows:

    For example, in the series of Xu et al. [43], comparing the diagnostic value of three RSS (i.e. ACR, EU-TIRADS and KSThR) in 2465 thyroid nodules, the rate of unnecessary FNA was lowest with the ACR-TIRADS (17.3%), followed by ETA-TIRADS (25.2%) and K-TIRADS (32.1%). Among nodules not submitted to FNA, 33.1 %, 37.7% and 38.2% thyroid cancers would be missed by the same TI-RADS, respectively. Finally, after applying adequate FNA cut-offs of each of these TI-RADS, 62.6%, 54.6% and 43.9% FNAC were avoided, respectively.

  24. 2 L 495: please insert comma behind RADS
    Done

  25. 2 L 509 ff: please rephrase
    The paragraph was rephrased as follows:

    However, this is at the cost of temporarily missing a significant proportion of carcinomas. Their diagnosis will be postponed until they eventually grow and are then diagnosed after they reach the cut-off threshold defined for FNA according to their US risk category. Most of the time, this strategy implies no significant loss of chance for the patient. However, the recommendation for no further evaluation, as specifically formulated in the ACR-TIRADS, could be considered with caution.

  26. 2 L521: 45 instead of forty-five;
    Done

  27. 4 L 582: please delete the second full stop
    Done

  28. 1 L 648: please delete REFERENCE
    Done

  29. 4 L 695f: please delete the square brackets
    Done

Reviewer 2 Report

I red carrefully the article entitled “the new era of TIRADs to stratify the risk of malignancy of thyroid nodules: stranghs, weakness and pitfalls”. It is a well written  review of the literature concerning all the features of the international TIRADs classifications.

The paper is well constructed, the first chapter describes the different Risk stratification system (RSS) existing in the field of the ultrasound of the thyroid nodules. For each RSS the categories and the US criterions are well described. The results of each RSS are presented according to the data of the literature.

The second chapter deals of the problem of the Cut offs of each RSS for FNA. The authors report for each RSS the diagnostic values after applying cut off and the problems of missed carcinomas.

The chapter 3 deals the weaknesses of each RSS, the 4 reports the cases that present diagnostic challenges for RSS. The chapter 5 explores the possibility of improvement for the RSS.

I think that this paper is a very interesting review of the literature concerning the stratification of the risk of malignancy of the thyroid nodules by US that is fundamental tool in the diagnois of throid carcinoma. It reports in a very rigorous manner the importance of the development of the RSS in the diagnosis of malignancy in the field of thyroid nodule. It allows to understand the importance of scientific contribution of the RSS in this field. It has the advantage to show that no RSS is perfect, and that clinical research must continue in US stratification to work to improve the US diagnosis of malignancy in thyroid nodules.

Furthermore, the paper has been written by a very expert team in the field.

I have some minor commentaries:

In chapter 1 :

-          the patterns of malignancy in Chilean-TIRADs should be described like in the other RSS.

-          In the table 1 : the abbreviation ETE for extra thyroidal extension should be defined  because the abbreviation is not defined in the 1st chapter.

-          In chapter 2 ,

o          Authors should add a table reporting cut off decision guidance for performing FNA for each RSS

o          the problem of missed carcinomas should be more discussed according to the prognosis of the differentiated thyroid carcinoma. A management should be proposed.

-          In chapter 5.4 : the chapter concerning MTC  Should be rewritten . The problem of the decision concerning the calcitonin level is more difficult than that is written. For high-risk cancer that will benefit for a thyroidectomy, the calcitonin dosage is not issue. The problem is more complex for moderate risk nodules. The literature and the experience show that for a calcitonin level over 100 pg/ml nodule larger than 1 cm are MTC. For levels below 100 ng/l and nodule larger than 1 cm the systematic calcitonin measurement does not bring a clear advantage for the diagnosis. Furthermore, the literature has showed that the systematic calcitonin measurement is not a cost-efficient method for the diagnosis of MTC.

Author Response

Dear reviewer,
thank you for your very interesting and kind comments.
We have tried to answer completely as follows:

In chapter 1 :

-          the patterns of malignancy in Chilean-TIRADs should be described like in the other RSS.
Thank you for the comment which is very logical. We were initially reluctant due to the complexity of the description in the report by Horvath at al.The following text was added:
TIRADS 2 corresponded to anechoic with hyperechoic spots, nonvascularized lesions, or to nonencapsulated, mixed isoechoic with hyperechoic spots lesions and to spongiform nodules. TIRADS 3 nodules were to hyper, iso, or hypoechoic, partially encapsulated nodules with peripheral vascularization, in Hashimoto’s thyroiditis. TIRADS 4A nodules were solid or mixed hyper, iso, or hypoechoic nodules, with a thin capsule, or hypoechoic lesion with ill-defined borders but without calcifications or hyper, iso, or hypoechoic, hypervascularized, encapsulated nodules with a thick capsule, containing calcifications (coarse or microcalcifications). TIRADS 4B corresponded to hypoechoic, nonencapsulated nodules, with irregular shape and margins, penetrating. vessels, with or without calcifications and TIRADS 5 to iso or hypoechoic, nonencapsulated nodules with multiple peripheral microcalcifications and hypervascularization or nonencapsulated, isoechoic mixed hypervascularized nodules with or without calcifications, without hyperechoic spots. 

-          In the table 1 : the abbreviation ETE for extra thyroidal extension should be defined  because the abbreviation is not defined in the 1st chapter.
Thank you for the comment. The meaning of ETE was added in Table 1.

-          In chapter 2 ,

o          Authors should add a table reporting cut off decision guidance for performing FNA for each RSS
Thank you for this comment. The following Table has been added.

RSS

TIRADS 2

or very low risk

TIRADS 3
or low risk

TIRADS 4
or intermediate risk

TIRADS 5
or high risk

Small nodules < 10 mm

Chilean
TIRADS

No FNA or follow-up

FNA (no cut-off) or follow-up

FNA
(no cut-off)

FNA
(no cut-off)

FNA if > 3-4mm and feasible

Kwak-TIRADS

No FNA

No FNA

TIRADS 4a : ≥ 25 mm
TIRADS 4B : 15mm

TIRADS 4 C  and 5: ≥ 10 mm

Issue not addressed

BTA

No FNA

All nodules

All nodules

All nodules

AACE

No FNA

≥ 20 mm and
growing lesion or risk factors

≥ 20 mm

≥ 10 mm

< 5mm no FNA
5-10mm FNA if clinical or US risk factors or PP

ATA

≥ 20 mm or
observation

≥ 15 mm

≥ 10 mm

≥ 10 mm

5-10mm FNA if clinical or US risk factors or PP

K-TIRADS

≥ 20 mm

≥ 15 mm

≥ 10 mm

≥ 10 mm

≥ 5 mm selective cases

EU-TIRADS

No FNA

> 20 mm

> 15 mm

> 10 mm

FNA or active surveillance, PP

ACR-TIRADS

No FNA

≥ 25 mm

≥ 15 mm

≥ 10 mm

Issue not addressed

C-TIRADS

No FNA

No FNA

≥ 15 mm

≥ 10 mm

US risk factors

o          the problem of missed carcinomas should be more discussed according to the prognosis of the differentiated thyroid carcinoma. A management should be proposed.
Thank you for this comment. The paragraph was completely rewritten as follows:

As a conclusion, all RSSs seem to reduce effectively the number of unnecessary FNAs. However, this is at the cost of temporarily missing a significant proportion of carcinomas. Their diagnosis will be postponed until they eventually grow and are then diagnosed after they reach the cut-off threshold defined for FNA according to their US risk category. Most of the time, this strategy implies no significant loss of chance for the patient. This is due to the statistical predominance of papillary carcinomas of low and intermediate risks among all thyroid cancers. However, looking for lymph node or extra-thyroidal extension, including clinical factors such as age, sex, personal and family history with risk factors of thyroid cancer, tumour growth rate and also serum calcitonin whenever judged relevant is critical for making the right decision to prevent missing more aggressive carcinomas . Thus, the recommendation for no further evaluation, as specifically formulated in the ACR-TIRADS, should be considered with caution and put into perspective including clinical and biological data.

-          In chapter 5.4 : the chapter concerning MTC  Should be rewritten . The problem of the decision concerning the calcitonin level is more difficult than that is written. For high-risk cancer that will benefit for a thyroidectomy, the calcitonin dosage is not issue. The problem is more complex for moderate risk nodules. The literature and the experience show that for a calcitonin level over 100 pg/ml nodule larger than 1 cm are MTC. For levels below 100 ng/l and nodule larger than 1 cm the systematic calcitonin measurement does not bring a clear advantage for the diagnosis. Furthermore, the literature has showed that the systematic calcitonin measurement is not a cost-efficient method for the diagnosis of MTC.
Thank you for this remark. We have added all your comments in the paragraph: 

Calcitonin may detect C-cell hyperplasia and medullary thyroid cancer (MTC). However, most guidelines cannot recommend for or against routine calcitonin measurement in patients with thyroid nodules. A recent review [69] demonstrated that calcitonin has good sensitivity and specificity to diagnose MTC and could be useful when available in the evaluation of thyroid nodules. The literature and the experience show that for a calcitonin level over 100 pg/ml nodule larger than 1 cm are MTC. For levels below 100 ng/l and that in nodules larger than 1 cm the systematic calcitonin measurement does not bring a clear advantage for the diagnosis, especially if at low or intermediate US risk. However the value of routine testing in patients with thyroid nodules remains questionable, due to the low prevalence of MTC and whether routine calcitonin testing improves prognosis in MTC patients remains unclear. In clinical practice, situations associated with false positivity of calcitonin tests (eg. renal insufficiency, treatment with proton pump inhibitor, obesity) and the correlation of calcitonin value with the nodule volume should be taken into account for the interpretation of the result. Calcitonin measurement remains mandatory in case active surveillance of EU-TIRADS 5 nodules or proven microcarcinomas is considered and before surgery or thermal ablation. Anyway, the heterogeneous US presentation of MTC [70] and the low sensitivity of FNA in detecting MTC [71] has to be taken into account during the clinical practice.